# Improving Analog Neural Network Robustness: A Noise-Agnostic Approach with Explainable Regularizations

**Alice Duque, Pedro Freire,**
Egor Manuylovich, Dmitrii Stoliarov, Jaroslaw Prilepsky, Sergei Turitsyn
Aston Institute of Photonic Technologies, Aston University, Birmingham, UK
`freiredp@aston.ac.uk`

## Abstract

This work tackles the critical challenge of mitigating "hardware noise" in deep analog neural networks, a major obstacle in advancing analog signal processing devices. We propose a comprehensive, hardware-agnostic solution to address both correlated and uncorrelated noise affecting the activation layers of deep neural models. The novelty of our approach lies in its ability to demystify the "black box" nature of noise-resilient networks by revealing the underlying mechanisms that reduce sensitivity to noise. In doing so, we introduce a new explainable regularization framework that harnesses these mechanisms to significantly enhance noise robustness in deep neural architectures, obtaining over 53% accuracy improvement in noisy environments, when compared to models with standard training.

## 1 Introduction

As the applications of artificial intelligence (AI) continue to surge, the computational power required to sustain this growth is leading to increasingly high energy consumption and a corresponding significant carbon footprint [1, 2]. This escalating environmental impact necessitates the exploration of innovative solutions to enhance computational efficiency [3]. Among these emerging approaches, analog neural networks have gained attention due to their potential to offer faster processing speeds with lower energy requirements, thereby addressing the critical need for sustainable AI development [4, 5].

However, analog circuits have a major drawback: they are naturally much more sensitive to noise compared to their digital counterparts. Noise in neural networks is not a particularly new topic of study, and extensive solutions have been proposed to overcome noisy inputs [6] and noisy labels [7],[8]. However, since the majority of networks in literature run on digital devices, these types of noise are external to the network itself, failing to address a type of noise that greatly affects analog networks in particular: hardware noise.

While it is well-recognized that hardware noise is a significant source of uncertainty in analog networks [9], it remains a relatively underexplored central topic. Ref [10] offers a broad analysis of noise effects across various architectures. However, the proposed solutions involve architectural modifications that result in increased hardware complexity. In contrast, [11] provides valuable insights into hardware noise but focuses on device-level solutions, which limit their general applicability. Ref. [12] demonstrates promising results by employing a device-agnostic approach during training. Nonetheless, that study lacks detailed explanations regarding the mechanisms behind the observed improvements.

In this work, we examine the impact of additive noise on feed-forward networks, along with the underlying mechanisms that contribute to the effectiveness of noise-aware training. In parallel, we

Submitted to the Second Workshop on Machine Learning with New Compute Paradigms at NeurIPS (MLNCP 2024). Do not distribute.

introduce a novel regularization strategy that enhances the network's robustness to correlated and uncorrelated hardware noise. This strategy not only offers a mathematically supported explanation, but it is also empirically validated through probability density function (PDF) analysis, providing strong justification for its effectiveness. We believe that the analysis of PDF evolution in neural networks introduced and applied here is a useful and important technique for designing and optimising their performance.

## 2   Noise in Neural Network Models

Building upon the classification proposed by Ref. [10], we categorize noise as either correlated or uncorrelated in this work. Noise is considered correlated when all neurons within a layer experience the same perturbation simultaneously, typically arising from variations in shared physical components, such as temperature, supply voltages, or laser sources. Conversely, uncorrelated noise refers to unique perturbations affecting each neuron independently, though these perturbations are drawn from the same underlying probability distribution. Consistent with other works in the literature, we model noise as being drawn from a zero-mean Gaussian distribution, with variance determined by the specific hardware characteristics.

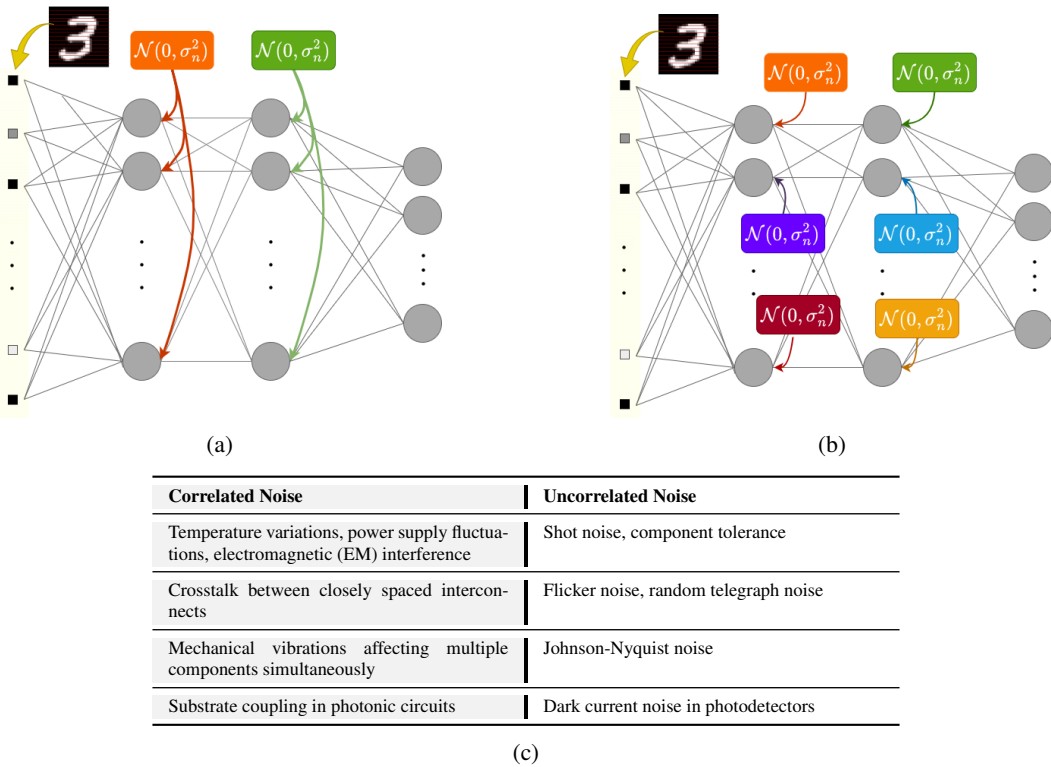

(a)                                                                           (b)

| Correlated Noise | Uncorrelated Noise |
|---|---|
| Temperature variations, power supply fluctuations, electromagnetic (EM) interference | Shot noise, component tolerance |
| Crosstalk between closely spaced interconnects | Flicker noise, random telegraph noise |
| Mechanical vibrations affecting multiple components simultaneously | Johnson-Nyquist noise |
| Substrate coupling in photonic circuits | Dark current noise in photodetectors |

(c)

Figure 1: Feed-forward network under (a) correlated and (b) uncorrelated noise. (c) Examples of such noise sources in analog NNs.

In this study, we focus on noise introduced after the activation functions (Figure 1) rather than within the weights. This is motivated by the fact that analog activation functions are in general more complex than passive connections and more likely to introduce noise. When noise is injected into a network trained to achieve high accuracy, a significant drop in performance is expected. This highlights the disparity between simulated and actual performance when these networks are deployed on noisy physical devices.

Among the solutions discussed in the literature, noise-aware training [12] assumes that neural networks can develop resilience to noise if trained with exposure to it. For that purpose, noise similar to what is expected during inference is injected during training, resulting in enhanced robustness. This method has shown promising results, leading to nearly a four-fold improvement in forecasting accuracy for a time-series prediction task. However, there is still a lack of understanding regarding

the mechanisms that drive these improvements. Specifically, it remains unclear how neural networks adjust to mitigate both correlated and uncorrelated additive noise so effectively. In the following sections, we present, to the best of our knowledge, for the first time, an analytical and mathematically explainable regularization strategy to achieve comparable noise mitigation results. Without loss of generality, we apply our approach to a fully connected multi-layer perceptron (MLP) with 2 hidden layers of 300 neurons each, with sigmoid as the activation function.

## 3  Designing Explainable Regularizers

Consider activations $a_j^{(l-1)}$ of $j^{\text{th}}$ neuron in layer $l-1$ are corrupted by noise $\hat{n}_j^{(l-1)} \sim \mathcal{N}(0, \sigma_n^2)$. In this scenario, these activations interact with weights $w_{ij}^{(l)}$ and biases $b_i^{(l)}$ to form pre-activations $z_i^{(l)}$:

$$z_i^{(l)} = \sum_j w_{ij}^{(l)}(a_j^{(l-1)} + \hat{n}_j^{(l-1)}) + b_i^{(l)}, \tag{1}$$

which can be written as

$$z_i^{(l)} = \sum_j w_{ij}^{(l)}a_j^{(l-1)} + b_i^{(l)} + \sum_j w_{ij}^{(l)}\hat{n}_j^{(l-1)}. \tag{2}$$

The last term corresponds to the total noise contribution on pre-activation $z_i^{(l)}$, or the *error factor*. If noise is correlated, then $\hat{n}_j = \hat{n}$ for all $j$, so the error factor becomes:

$$\sum_j w_{ij}^{(l)}\hat{n}_j^{(l-1)} = \hat{n}^{(l-1)} \sum_j w_{ij}^{(l)}. \tag{3}$$

This means that the error factor can be effectively eliminated if we make $\sum_j w_{ij}^{(l)} = 0$. In other words, correlated noise can be mitigated by ensuring that each row of the weight matrices sums to zero. This allows us to formulate the first explainable regularizer for additive correlated noise, which enforces this mathematical condition. By introducing a regularization term into the loss function during training, we can effectively impose this constraint, leading to improved noise robustness. The modified loss function becomes:

$$\text{Loss}_\theta = \mathcal{L}(y_{\text{pred}}, y_{\text{exp}}) + \sum_{l=2}^{L} \lambda_l \sum_i \left| \sum_j w_{ij}^{(l)} \right|. \tag{4}$$

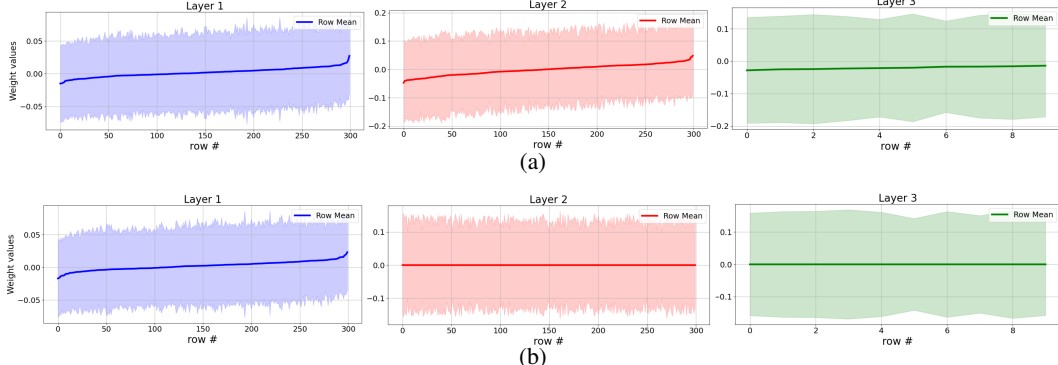

Figure 2: Per-row visualization of each weight matrix in a network with (a) standard training and (b) proposed regularization method. Regularization manages to push rows means toward zero to meet correlated noise mitigation strategy.

For visual inspection, Fig. 2 illustrates the effects of regularization on the distribution of weights per row, compared to the original neural network model trained without noise. The graphs depict the mean (solid line) and standard deviation (shaded areas) of each row within the weight matrices, with layers ordered by ascending mean values. As predicted by Eq. 4, all rows—except those of the input

weight matrix—have their mean values driven toward zero. This exception arises because noise only affects activations, meaning the input weights do not directly interact with it. In the following section, we will present evidence demonstrating the improvement in accuracy achieved through this straightforward regularization strategy.

For uncorrelated noise, canceling the error factor becomes more complex. Instead of attempting to eliminate it entirely, we aim to mitigate its propagation to subsequent layers. Given that all individual noise contributions within layer $l-1$ have zero mean Gaussian PDF, the error factor is also Gaussian with zero mean. Furthermore, since for a known $a_j^{(l-1)}$ the first two terms of equation 2 are deterministic, the pre-activation $z_i^{(l)}$ can be interpreted as a random Gaussian variable $\hat{z}_i^{(l)} \sim \mathcal{N}(\mu_{ij}^{(l)}, \sigma_n^2)$, whose mean is:

$$\mu_{ij}^{(l)} = \sum_j w_{ij}^{(l)} a_j^{(l-1)} + b_i^{(l)}. \tag{5}$$

From the known pdf $p_z(z)$ of $\hat{z}_i^{(l)}$, and given that $f(\cdot)$ is monotonic, one can derive the PDF $p_a(a)$ of $a_i^{(l)} = f(\hat{z}_i^{(l)})$ from [13] as:

$$p_a(a) = \frac{p_z[f^{-1}(a)]}{|\frac{da}{dz}|}, \tag{6}$$

where $f^{-1}(\cdot)$ is the inverse of activation function.

Fig. 3 a) shows the PDF $p_a(a)$, where $f(\cdot)$ is the sigmoid function, and given $\hat{z}_i^{(l)} \sim \mathcal{N}(\mu_{ij}^{(l)}, 0.2)$ for different values of $\mu_{ij}^{(l)}$. It's observed that input distributions centered further from the origin lead to output distributions of much-reduced variance. Intuitively, this occurs because noise fluctuations in the saturated regions of the activation function (where the derivative is close to zero) are largely suppressed and not transmitted to the output. Thus, in analog neural networks, if a set of weights and biases can shift the operating point of pre-activations toward these saturated regions, noise-induced perturbations are less likely to propagate to subsequent layers. This mechanism effectively reduces the impact of uncorrelated noise, enhancing the network's robustness to such perturbations. This also can give some design ideas for the activation function shape when this freedom is available.

It is important to point out that, in the output layer, no additional nonlinearities are available to counteract noise contributions, which will inevitably affect the readout values. However, we found that this impact can be mitigated by reducing the magnitude of the output weights. In typical scenarios, reducing weights alone does not enhance noise performance, as both the signal and noise are attenuated equally, resulting in no real improvement in the SNR. However, if the intermediate layers meet the condition where pre-activations operate predominantly in saturated regions, the activation values will also be highly saturated. In our case, which uses the sigmoid activation function, this implies that the majority of activation values will be either 0 or 1. In such a distribution, reducing the output weights does not lead to information loss for activations that are zero—these values constitute a significant portion of the activations in the saturated regime. Thus, noise contributions are reduced without loss of information from the zero-valued activations.

Moreover, for non-zero activations, which are at the maximum possible value (1 in the case of sigmoid), the output remains resilient to noise. This dual effect—noise suppression for zero activations and noise insensitivity for saturated ones—explains how reducing output weights, when combined with saturated intermediate layers, can mitigate the impact of noise on the final output. We can incorporate these conditions into our training by including new regularizer terms into our loss function, which becomes:

$$\text{Loss}_\theta = \mathcal{L}(y_{\text{pred}}, y_{\text{exp}}) + \lambda_1 \sum_i f'(z_i^{(2)}) + \sum_{l=2,3} \lambda_l \sum_i \sum_j (w_{ij}^{(l)})^2 \tag{7}$$

in which $f'(z_i)$ is the activation function's derivative at $z_i$ and $\lambda_i$ are tunable hyperparameters.

The first regularization term penalizes pre-activations that fall within regions of high derivative in the activation function, while the second term penalizes large weight magnitudes in all layers except the input layer. The constraint on the hidden layer weights addresses a practical issue encountered

during training. When we enforced pre-activations to move away from the origin, the network initially complied by amplifying the weights in the hidden layers. Although this approach effectively shifted the operating point toward the saturated region, it inadvertently amplified noise contributions. This increased noise then "leaked" into the linear region of the activation function, propagating through subsequent layers and negating any performance improvement.

To prevent this, we apply a light constraint on the hidden layer weights, ensuring that the network does not resort to this suboptimal strategy during training. Fig. 3 b), c), and d) illustrate the distributions of the pre-activations, post-activations, and output weights in the original model, the noise-aware trained model, and the regularized model. These visualizations confirm our hypothesis: the regularized model exhibits a distribution similar to that of the noise-aware model, but with a more pronounced shift toward the non-linear regions of the activation function.

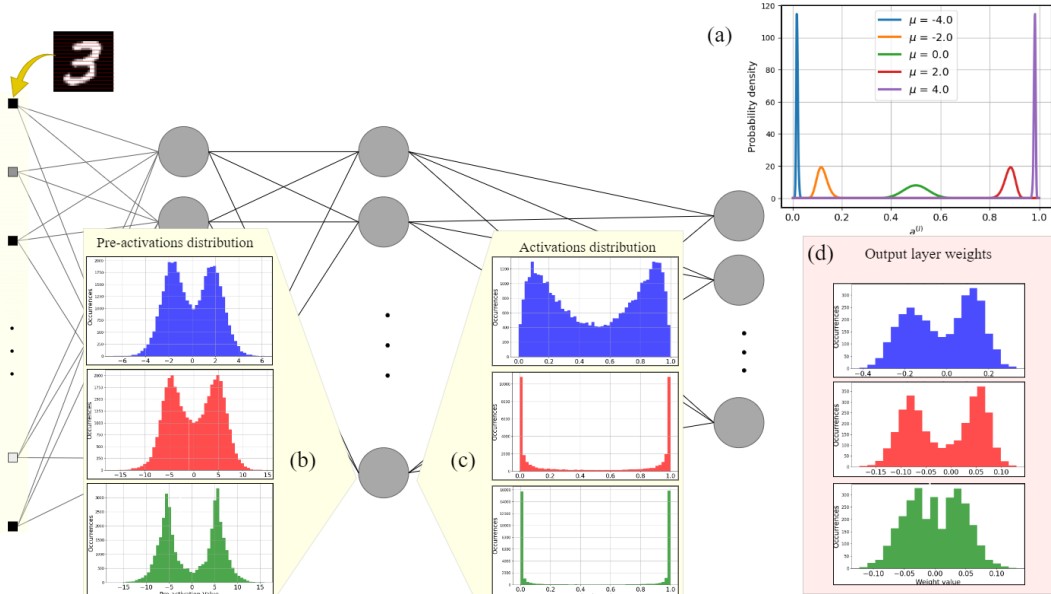

Figure 3: (a) PDF of random variable $\hat{z} \sim \mathcal{N}(\mu, 0.2)$ after undergoing sigmoid transformation, for different values of $\mu$. (b) Pre-activations, (c) post-activations and (d) output weights distributions for networks with different training methods. Blue: network with standard training; Red: network with noise-aware training; Green: network with proposed regularization method.

## 4   Results and discussion

Having established the explainability of the proposed regularization, we now demonstrate that the key features of noise-resilient networks are successfully incorporated by our regularization techniques, yielding significant performance improvements. To validate these claims and assess the overall performance, we evaluate the regularized network on three computer vision tasks, using the MNIST [14], Fashion MNIST [15] and QuickDraw [16] datasets. The results highlight the efficacy of the regularization in enhancing noise robustness while maintaining strong task performance. For the sake of reproducibility, all code used in these experiments is available in [17] .

In all tasks, the application of regularization effectively eliminated the impact of correlated noise, as shown in Fig. 4a)-(c), resulting in a flat accuracy curve with no observable performance degradation. For uncorrelated noise, substantial improvements were also achieved. On the MNIST dataset, the network experienced only a 4.77% drop in accuracy when subjected to hardware noise with variance as high as $\sigma^2 = 1.0$, compared to its performance in a noiseless environment. By contrast, a standard network trained without any noise-aware techniques showed a dramatic 41.52% accuracy decline under the same conditions. Similarly, in the more challenging Fashion MNIST task, regularization reduced the accuracy drop to 7.22%, compared to 53.86% for the standard trained network. For QuickDraw, the accuracy drop went from 65.58% to 35.29%, when the proposed regularization was in place. It's worth mentioning that 20 classes were selected from the original QuickDraw dataset – twice as many as MNIST and Fashion MNIST –, which reflect in the poorer performance

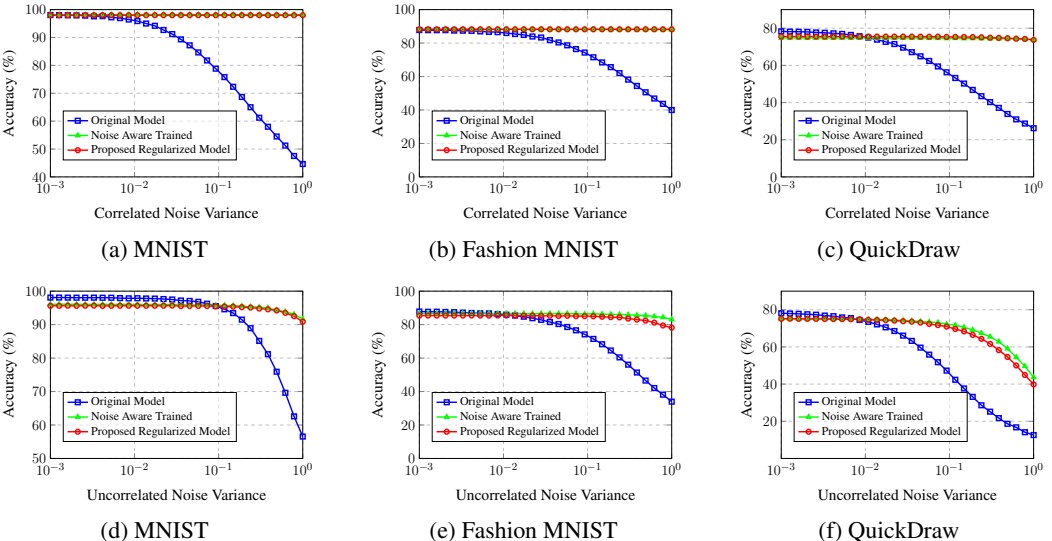

Figure 4: Performance comparison between models with standard training, noise-aware training and the proposed regularized model, for MNIST, Fashion MNIST, and QuickDraw datasets.

comparatively. Nonetheless, these results demonstrate a significant enhancement in noise robustness. However, this improvement comes with a slight penalty to the maximum achievable accuracy. For MNIST, a network with standard training achieved 98.06% accuracy in a noise-free environment, while the regularized network reached 95.62% under the same conditions. Similarly, for Fashion MNIST, the accuracy for the standard network was 87.78%, while the regularized network achieved 85.37%. For QuickDraw, the figures are 78.11% and 75.12%. It is important to note that this accuracy reduction is also observed in networks trained with noise exposure.

The promising results from the treatment of correlated noise indicate that the imposed weight constraint is relatively mild, allowing the network to assimilate it without compromising overall performance. In contrast, the regularization designed for uncorrelated noise introduces a more stringent constraint, which imposes a notable trade-off during training and can limit the maximum achievable accuracy. Despite this, the overall performance improvements underscore the effectiveness of the method. However, a key challenge of the proposed approach lies in the difficulty of optimizing the training process. The custom regularization terms require a systematic and often repetitive search for the optimal hyperparameters $\lambda_i$, typically through a trial-and-error process. This adds significant complexity to the training pipeline, substantially increasing both the training time and computational cost. Fine-tuning these regularization coefficients is critical for balancing the trade-offs between noise suppression and accuracy, but it also represents a bottleneck in terms of scalability, especially in larger models or more complex architectures.

## 5 Conclusion

In this work, we explored the fundamental mechanisms that allow a neural network to achieve resilience to hardware noise without introducing additional architectural complexity, relying solely on optimized weight and bias distributions. Our novel approach, based on the analysis of the pdf evolution throughout the network, offers a valuable design and optimization tool for improving noise robustness. By presenting explainable regularization techniques, we demonstrated an over 53% improvement in accuracy under high levels of hardware noise, underscoring the effectiveness of the method. Importantly, the proposed technique is device-agnostic, offering broad applicability across various analog neural networks affected by additive noise. This generalizability, combined with the simplicity of implementation, positions our approach as a significant advancement for the design of high-performance, noise-resilient neural networks in real-world hardware applications.

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
