# OpenReview forum: "Improving Analog Neural Network Robustness: A Noise-Agnostic Approach with Explainable Regularizations"
_NeurIPS.cc/2024/Workshop/MLNCP — MLNCP Poster_

### Official Review · Reviewer_cQT2 · 2024-09-27
**A simple, yet interesting study of possible regularizations to make networks more noise robust**

**Rating:** 7
**Confidence:** 3

**Review:**

This submission explores possible regularizations to reduce the influence of gaussian activation noise on the network outputs. The authors give expressions for correlated noise as well as uncorrelated noise. Noise robustness is an important property of neural networks if they are to be run on analog, quantum, or other volatile systems.

The aim of the paper is to replace the need to train neural networks (assuming they are trained digitally) explicitly with noise which would require detailed knowledge of noise strength and correlation by introducing regularization terms. Another point of view of what this paper tries to achieve is that it seeks to explain what actually happens to the weights when networks are trained under noise. This explanatory perspective, however, is only empirically backed, not shown in detail or as direct comparison between noise-trained and regularization-trained networks.

For the case of correlated noise, the authors find a simple way to eliminate the influence of the noise which holds up in experiments.
Furthermore, for the case of uncorrelated noise, networks trained with these regularizations still experience a significant drop (4.77% for MNIST, 7.2% for Fashion-MNIST) in accuracy for strong noise. This stands in contrast to networks that are trained under noise which might still be the preferred option for high noise variances (if a detailed noise model is available).

The proposed mechanisms that networks can resort to in order to deal with uncorrelated noise such as driving the signal into the constant part of the non-linearity likely explains the overall lower achievable accuracy on MNIST/Fashion-MNIST.

The experimental section of the paper could be more extensive. The theory could be extended to transformers where activations in queries and keys are multiplied, leading to potentially different noise distributions in the resulting variables. I believe the scope is sufficient for a workshop.

---

### Official Review · Reviewer_SQg4 · 2024-10-01
**Interesting explanation but no gain compared to state of the art**

**Rating:** 4
**Confidence:** 3

**Review:**

The authors propose a new regularization function to reduce the impact of the activation noise and explain why it works. Zeroing the sum of row weights for correlated noise completely clears the noise. For uncorrelated noise, the authors attempt to concentrate the preactivation values in the saturated part of the activation function.

Results show a good impact on the accuracy compared to an approach with standard training. Nevertheless, a "noise-aware" training gives even better results. It is not completely clear what the authors mean by "noise-aware" training. Since the results seem better than the proposed regularization function, authors should provide details about their test bench and why they claim a contribution.

Here are a few other things that could enhance the paper :
1) Evaluate the proposed method on networks and datasets that are more complex than MNIST and MNIST fashion.
2) Double-check equation (7). It is unclear why you only seem to use the preactivation values of layer two, but you consider the square weight values of all layers.
3) Your proposed method for uncorrelated noise and Sharpness-Aware-Minimization (SAM) probably share some similarities since you want the activations to be in a low derivative region. Please discuss this.

---

### Decision · Program_Chairs · 2024-10-10

Accept (Poster)